# Tinnitus and Traumatic Memory

**DOI:** 10.3390/brainsci12111585

**Published:** 2022-11-20

**Authors:** Marc Fagelson

**Affiliations:** 1Department of Audiology & Speech-Language Pathology, East Tennessee State University, Johnson City, TN 37614, USA; fagelson@etsu.edu; 2Audiology Section, Mountain Home Veterans’ Medical Center, Johnson City, TN 37684, USA

**Keywords:** tinnitus, posttraumatic stress disorder, cognitive-behavioral therapy, emotional memory

## Abstract

Events linked to post-traumatic stress disorder (PTSD) influence psychological and physical health through the generation, exacerbation, and maintenance of symptoms such as anxiety, hyperarousal, and avoidance. Depending upon circumstance, traumatic events may also contribute to the onset of tinnitus, post-traumatic headache, and memory problems. PTSD should be considered a psychological injury, andwhile tinnitus is a symptom, its onset and sound quality may be connected in memory to the injury, thereby evincingthe capacity to exacerbate the trauma’s effects. The myriad attributes, psychological and mechanistic, shared by tinnitus and PTSD offer tinnitus investigators the opportunity to draw from the rich and long-practiced strategies implemented for trauma counseling. Mechanisms and interventions understood through the lens of traumatic exposures may inform the clinical management of tinnitus disorder, and future studies may assess the effect of PTSD intervention on co-occurring conditions. This brief summary considered literature from both the hearing and trauma disciplines, with the goal of reviewing mechanisms shared between tinnitus and PTSD, as well as clinical reports supporting mutual reinforcement of both their symptoms and the effects of therapeutic approaches.

## 1. Introduction

### 1.1. Subjective Tinnitus

Most references to tinnitus center on the subjective experience of an auditory perception in the absence of sound external to the hearer [1]. Subjective tinnitus may arise from a variety of insults to the auditory system, including excessive sound pressure levels, environmental toxins, ototoxic medications, and pathologies including Meniere’s Disease and vestibular schwannoma. Although damage to the auditory mechanism on its own may produce tinnitus, it is also the case that tinnitus-related neural activity may emerge as a consequence of central adaptations to the aforementioned pathologies. Roberts [2] offered a review of auditory pathway activity associated with tinnitus appearance and suppression. Such activity included burst-like behavior across arrays of neurons, changes to spontaneous activity throughout the pathway, and the appearance of unusual electrophysiologic findings such as normal ABR wave V latencies even in the presence of delayed or absent earlier waves. For example, exposure to damaging sound pressure levels produced durable changes to the growth functions of neurons in the central pathway [3]. The changes manifested in animals’ cochlear nuclei and inferior colliculi neurons as decreased thresholds to electrical stimulation following noise exposures; the central pathway appeared to employ some combination of auditory gain and/or central disinhibition [4]. While there may be putative value related to the pathway’s increasing central activity in response to peripheral damage, the by-products of such changes may not be beneficial, as tinnitus and disorders of sound tolerance may result from the central adaptations. Therefore, tinnitus may emerge as a primary effect of traumatizing exposures, or as a secondary effect of the central pathway’s compensatory adjustment to such damage. The diverse causes of tinnitus-associated damage, as well as the myriad and complex central mechanisms that may contribute to its detection and severity, leave patients and practitioners in the current sub-optimal situation in which a patient’s experience with tinnitus may be as unique as it is uncurable.

### 1.2. Tinnitus and Trauma

Additional contributions to tinnitus severity may involve factors unrelated to the sensation of hearing, and oftentimes the greater the relative contribution of non-auditory influences, the worse the tinnitus. Links between tinnitus and mental health status were reported, albeit using different terminology, more than 2500 years ago [5]. The potential for tinnitus and mental health status to reinforce one another in powerful and negative ways was obvious to patients and an interest of researchers for several decades [6,7,8,9,10,11,12]. Studies relating traumatic exposures to tinnitus suggested that memories and trauma-related hyperarousal conspired to exacerbate not only tinnitus distress but related sound tolerance issues [7,10,13,14]. Traumatic onset was specified by Kreuzer et al. [15] as exacerbating tinnitus “burden”. A sudden traumatic onset and potential for catastrophic associations related to such tinnitus onset may contribute to durable symptoms arising from the combination of negative influences. As Janet suggested, victims of trauma suffer “mainly from reminiscences” of the event [16]. One lesson learned from trauma psychology: when considering the co-occurrence of tinnitus and PTSD, the patient’s ability to manage tinnitus may rely at least in part upon recognition of the events and related memories that the patient associates with the traumatic experience. Further it is possible that the tinnitus reinforces such “reminiscences” through its constancy and associations with the trauma [10,11].

Distinctions between “tinnitus” and “tinnitus disorder,” and their relation to non-auditory contributors to tinnitus, were reinforced as researchers and clinicians reported links between tinnitus and emotion, mental health status, insomnia, trauma, and other factors [17]. Sound sources, or sound qualities annoying to one person may not be annoying to another, and the likely difference in annoyance would be due at least in part to the value ascribed to the sound by the patient [18]. Auditory events that triggered unwanted thoughts and memories likely would be processed with greater vigor and urgency than sounds evaluated as neutral and innocuous; however, a patient misinterpreting a sound’s value may employ such processing unnecessarily, and with negative consequences. Factors related to tinnitus salience—its onset, its resemblance to sounds experienced during traumatic episodes—reportedly contributed to the amount of distress the tinnitus produced as well as to the severity of co-occurring traumatic mental health injury [10,11,13].

### 1.3. Post-traumatic Stress Disorder

The galaxy of symptoms that may result from exposure to, perpetration of, or witnessing of violence and trauma were described at least as long ago as Homer’s Iliad and Odyssey [19]. Military and civilian trauma, captivity, and abuse in its many forms led to the discovery and eventual diagnostic criteria for PTSD [15,20]. Early psychoanalysts attributed symptom severity and durability to, among other things, traumatic reminders or, as Janet suggested, “reminiscences” [21]. The sensory scene associated with the trauma may support the persistence of traumatic memories, as can injuries sustained during the exposure. The links between exposures and subsequent symptoms were apparent centuries ago [19]. Given the likelihood that violent exposures include high sound pressure levels, head trauma, and toxins in the environment, there would be a strong possibility that sensory disorders related to hearing would result from exposures that might cause PTSD [14,22]. Here, we consider the consequences of tinnitus playing the part of an aversive auditory event and memory derived from trauma.

PTSD was first cited by name in the third edition (1986) of the Diagnostic Statistical Manual of the American Psychiatric Association [23]. The severity of the symptoms endorsing the diagnosis compelled investigators to consider PTSD a psychological injury rather than a disorder [19]. As conceptualized by Shay and others, notably Judith Herman [16,21], exposures that produced PTSD affected a person’s character fundamentally; the experience distorted or destroyed that person’s understanding of the moral order and imperatives accepted and with other individuals who comprised their societal connections. Herman [16] stated: “Every instance of severe traumatic psychological injury is a standing challenge to the rightness of the social order”.

PTSD affects hundreds of millions of people across the globe. Hoppen & Morina [24], reporting specifically on war-exposed individuals, estimated “that about 1.45 billion individuals worldwide have experienced war between 1989 and 2015 and were still alive in 2015, including one billion adults”. Their meta-analysis estimated approximately 354 million adults who experienced war experienced PTSD symptoms and/or major depression. They also estimated that about 117 million suffered from comorbid PTSD and major depression. Koenen et al. [25] reported findings from the World Mental Health Surveys, and of 123,299 participants across 26 nations, 71,083 reported histories of trauma and related PTSD symptoms of varying in severity and duration. Fewer than 50% of respondents reporting PTSD symptoms indicated seeking clinical services, and the authors pointed out that, in the US, where clinical intervention was most readily available, about half the respondents reported seeking help; of those, only 58% received services from a mental health practitioner. As it is acknowledged that tinnitus may be underreported [25], the prevalence of tinnitus and PTSD co-occurring across the general population remains difficult to estimate with certainty.

The National Center for PTSD (NCPTSD) in the US estimated that approximately 6% of all adults experienced PTSD symptoms at some point in their lives [26]. The prevalence among females was twice that of males (8% vs. 4%). Nearly 15 million adults received a new diagnosis of PTSD annually, however the NCPTSD asserted that the worldwide value represented a fraction of the number of people who experienced a traumatic event that could cause PTSD. In the Veteran’s Affairs system in the US, patients may be provided service connection, and related compensation, for injury and disease linked to military service. In the 2021 veteran’s benefit report [27], the number of compensations for tinnitus (2.5 million) exceeded all other service-related conditions, while hearing loss cases ranked second (1.4 million), and PTSD compensations ranked fourth (1.3 million patients). In 2020, the number of patients receiving compensation for tinnitus again led all other conditions (nearly 150,000 new cases), while new service connections for PTSD increased at the 8th highest rate (45,153 new cases).

Cima et al.’s European Guideline specified PTSD as a “frequent comorbidity” in the “Reaction to severe stress and adjustment disorders” category [28]. The high co-occurrence of PTSD and tinnitus underscores the likelihood that the two share many mechanisms of action related to onset, symptom severity, and symptom persistence. Note also that Shay [19] and Lifton [29] indicated that “instances of trauma” were not limited to the victims of trauma but rather included the perpetrators and observers of trauma. Their studies and fieldwork interviewing patients affirmed that people did not have to be physically traumatized to develop PTSD. Hinton et al. [10,11] further acknowledged the role of tinnitus as increasing PTSD prevalence and amplifying symptom severity.

## 2. Methods

Article selection for this brief review did not adhere to a formal protocol. The intent of this piece was to report on shared mechanisms and interventions specified in the tinnitus and trauma literatures. A systematic review would doubtless reveal additional references and would provide a reasonable next step toward elucidating relevant associations between tinnitus and PTSD.

### 2.1. Shared Mechanisms: Hearing

For most people, sound, speech, music, and sirens can transport and transform. In the context of tinnitus linked to trauma, a sound’s interpretation may take on elements of a threatening or sinister situation in which the patient endured physical and psychological injury. The duration of the negative event, the injuries, the level of threat, and any subsequent repetition or re-experiencing of the trauma would contribute to symptoms consistent with those of PTSD. Studies by Hinton and colleagues [10,11] included patient narratives in which a tinnitus sound that resembled a specific element of a traumatic exposure produced “catastrophic associations” that exacerbated the severity of PTSD symptoms. One specific example centered on insect-like tinnitus that reminded observers of the ubiquitous buzzing sound of insects they experienced navigating their streets and towns amidst a 20% mortality rate among Cambodian civilians from 1975–1979.

Evidence from both the trauma and audiologic literature supports mutual reinforcement between the sequalae of traumatic events, their associations, and tinnitus distress. The term “auditory exclusion” refers to an individual’s inability to detect or focus on obvious acoustic events in the environment. Auditory exclusion was reported by Moore et al. [30] as a perceptual event that may influence, during periods of high stress, an individual’s ability to accurately process the acoustic environment. Reports from both law enforcement and combat theaters indicated that during episodes of high stress, for example, threatening environments, individuals may not recall firing their own gun, or other high-level sounds essential to making sense of a confusing or threatening environment.

Although linked to traumatic exposures, auditory exclusion was compared to tunnel vision as an unusual sensory processing event in which the perceiver would focus exclusively on elements associated with a traumatic exposure at the expense of attention being available for the rest of the environment. Bremner [31] reported visual exclusion as well as auditory exclusion in patients whose trauma histories included exposures on 9/11. The misinterpretation of sensory environments was a characteristic associated with traumatic exposures, amplified by the finding that obvious environmental features were reportedly undetected by a population of trauma survivors. Such reports underscored the effects of trauma on the survivor, as an individual’s focus on traumatic events or reminders may render other sensory events, even powerful and important elements of a scene, undetected. Suppose in such a case that tinnitus arose during the event; for some individuals, the tinnitus signal might take priority, often if not always, over other salient environmental features. Tinnitus would in such cases be assured a place of prominence in the patient’s sensory world.

The value assigned to a familiar sound is not immutable. Consider that sounds once bothersome can over time become less intrusive; noise from the street or nearby railway that keeps one awake at night usually loses that effect after several days of exposure. The opposite is also true; a sound that at one time was innocuous can become bothersome as a result of evolving context and experience; one might enjoy the sound of a steady rainfall until the individual buys a home in an area that floods easily, at which time the steady rain might take on a different value.

Although the senses serve and support the ecological needs of the perceiver, some sensory experiences evoke strong emotions that may not be beneficial to the perceiver. A sound linked to a joyful event will carry the value associated with the event just as a sound linked to trauma will carry the value associated with the trauma. Inserted into this malleable and dynamic processing system, tinnitus has the potential to influence a patient’s emotional state in a variety of circumstances, typically amplifying negative thoughts while negating positive thoughts. Tinnitus resulting from a specific event, particularly a traumatic event, may sustain patients’ responses and symptom severity in a manner that differs substantially from tinnitus that appeared and exerted its influence in a gradual way over time [13,15]. Mutual reinforcement between tinnitus and PTSD is one of the more powerful examples of durable links between mental health status and tinnitus severity.

### 2.2. Shared Mechanisms: Psychological

As is the case with susceptibility to noise induced tinnitus, several predisposing and/or co-occurring conditions may increase the likelihood that an individual develops PTSD after a traumatic exposure [21,32]. The relation between traumatic brain injury (TBI), and PTSD remains a topic of debate, as the effects of one may be difficult to distinguish from the effects of the other. Despite substantial variability across studies, PTSD and TBI co-occur routinely in blast-exposed soldiers and veterans, as well as civilian victims of motor vehicle accidents [33]. Hoge et al. [34] interviewed 2525 US veterans returning from Iraq and reported PTSD among nearly half the veterans who reported injuries with loss of consciousness (124, or 4.9% of the entire sample). Additionally, 260 (10.3%) sustained injuries that produced “altered mental status,” 27.3% of whom experienced symptoms of PTSD. Although distinguishing effects and mechanisms of PTSD and TBI may remain a challenge, the likelihood of their co-occurring with, and influencing tinnitus is unfortunately common. In a large veteran’s hospital in the US, the rate of PTSD among patients seen for tinnitus-related services exceeded 35% [13], a trend unchanged across more than 20 years of practice and 1300 patients.

Reviews of neural mechanisms associated with trauma and chronic stress experienced by soldiers and veterans identified several areas of overlap between PTSD and tinnitus [13,33]. Of specific interest, neural activity associated with signal salience, and the response to signals deemed salient, has the potential to influence symptom severity for both tinnitus and PTSD. Bremner’s [35] imaging studies revealed the effects of stress during investigation of the hippocampus, amygdala, and anterior insular cortex, among many other areas. Chronic trauma-related elevated levels of stress hormones co-occurred with impaired regions of the hippocampus to the extent that MRI revealed reductions in hippocampal volume among combat-exposed veterans [31,35]. By influencing activity in these areas, traumatic exposures and their sequalae contributed to the chronic hyperarousal, negative emotions, and exaggerated startle responses suggested by the PTSD diagnostic criteria. Other consequences of traumatic exposures may include mislabeling of sensory events, ref. [36] thereby contributing to inappropriate startle responses, and fostering avoidance in patients.

Poe et al. [37] suggested an additional structure, namely the locus coeruleus (LC), whose activity may bear consideration with regard to the effect a sound has on arousal. The LC is comprised of cells containing noradrenalin (NA), and its connections throughout the brain ensure its activity likely plays a “critical role in core physiologic and behavioral processes” [37]. Few studies report recordings from the LC, as it is a small nucleus deep in the brainstem, and understanding of the correlations between its activity and external events remains limited. However, novel approaches investigating LC activity revealed a relation between LC activity and adaptations to behavior that emerged when animals were exposed to sounds paired with noxious stimulation.

An investigation by Martins & Froemke [38] demonstrated durable changes in LC activity that were associated with changes in sensitivity to sounds as represented in the primary auditory cortex when sounds were paired with electric shocks. After isolating LC outputs, they found far stronger LC responses to tail pinch than to presentation of an innocuous signal, such as a puretone. When such “innocuous” tones were paired with shocks, the “LC neurons developed and maintained auditory responses afterwards. Locus coeruleus plasticity induced changes in A1 responses lasting at least hours and improved auditory perception for days to weeks” (p. 1483). Poe et al. [37] suggested that because resources required for plasticity were finite, neural circuits competed for the metabolic and molecular requirements with the outcome that the most salient stimuli would “win” the required resources. Some inputs will be favored as elements of learning, memory, and arousal while other inputs, those deemed of less salience or value, would be suppressed. As the LC and its associated circuitry influenced, if not at times dominated, distribution of NA throughout cortical and subcortical regions, stress-induced changes in NA levels, such as those identified by Bremner [31], underscored the potential value of the LC as a contributor to the hyperarousal symptoms. Further, the LC’s demonstration of plasticity in the face of stress-related sound experiences may contribute to trauma-related tinnitus disorder as well. Certainly, the LC’s putative contribution could be particularly strong when tinnitus onset and/or tinnitus sound takes on the added salience associated with a traumatic reminder.

### 2.3. PTSD and Tinnitus Disorder

While DeRidder et al. [17] linked definitions of tinnitus and tinnitus disorder to DSM criteria for somatic symptom disorder, a similar strategy can be employed regarding putative alignments between tinnitus and PTSD. The DSM criteria specified for PTSD diagnosis share elements with varying degrees to items specified by DeRidder et al. [17] as endorsing tinnitus disorder diagnosis. Tinnitus earned its reputation as an exacerbator of mental health distress, and its effects may be most obvious when viewed in the context of memories and symptoms associated with injury such as PTSD. A brief review of the criteria for PTSD from the DSM 5th Edition [39] and their relation to elements of tinnitus disorder follows.

Criterion A specifies “Exposure to actual or threatened death, serious injury, or sexual violence” without necessarily requiring excessive sound level, head trauma, or other events associated specifically with hearing or tinnitus. It is worth considering the influence a low-SPL event could exert on tinnitus that preceded the event. Negative emotional memories could increase tinnitus salience over time, particularly if the tinnitus sound resembled a trauma reminder. Changes to the tinnitus effects would therefore be susceptible to trauma effects independent of toxic sound levels.

Criterion B specifies, “Presence of intrusion symptoms associated with traumatic event(s)” such as “recurrent memories of event, dreams, flashbacks, intense psychological distress re: cues that symbolize or represent the event”. Hinton et al. [10] questioned trauma survivors regarding the tinnitus sound serving as a trauma reminder. Across participants, several tinnitus sounds producing “traumatic reminders” and “catastrophic associations” were identified; these sounds were specific to instances of trauma, and reported by participants as triggers consistent with Criterion B.

Criterion C specifies behavior at which many patients with bothersome tinnitus would no doubt scoff: “Persistent avoidance of stimuli associated with the traumatic event(s), beginning after the traumatic event(s) occurred”. While a trauma victim may isolate as a way to feel safe, no such option exists in tinnitus cases (i.e., the patient reports that there is no escape from tinnitus), although some patients take note of certain situations or environments in which tinnitus worsens. In this sense, cataloging or maintaining a diary related to tinnitus-related observations may support the patient in successfully navigating related or emerging environmental challenges. Regardless, a patient’s perceived or in some cases compulsive need to avoid tinnitus may worsen the experience as the sensation persists over time.

Criterion D specifies the influence of a traumatic exposure on emotions and ongoing thoughts about the event: “Negative alterations in cognitions and mood associated with the traumatic event(s), beginning or worsening after the traumatic event(s) occurred”. Again, although a patient thus affected may not experience tinnitus, clinicians should be mindful that co-occurring tinnitus may reinforce negative PTSD-related thoughts and emotions, at least with regard to its resemblance to sounds associated with trauma [10,11]. The DSM criteria further indicate that criterion D may include persistent negative emotional state, disengagement from friends and family, and negative beliefs about acquaintances and the world that contribute to withdrawal from social situations. Each of the items endorsing criterion D could reduce patients’ willingness to remain active socially, thereby minimizing the potential benefits associated with exposure to external sounds and opportunities for engaging activities or conversation.

Criterion E specifies, “Marked alterations in arousal and reactivity associated with the traumatic event(s), beginning or worsening after the traumatic event(s) occurred”. The DSM [39] provides examples of arousal symptoms, some of which are questioned on several tinnitus intake forms, including irritability and anger with minimal provocation. Additional forms of hypervigilance with relevance for patients with tinnitus may include sound intolerance and exaggerated startle response, concentration problems, sleep disturbance (e.g., difficulty falling or staying asleep or restless sleep). That several items included in Criterion E would be assessed during routine tinnitus evaluation should serve as a reminder that measures of symptom severity, in addition to the aforementioned overlap between mechanisms, demonstrate the strength of the association between tinnitus and PTSD. In a large clinical cohort, Fagelson [13] reported that patients with PTSD were four times more likely than those without PTSD to report reactive tinnitus. The patients with PTSD were twice as likely to report that sound tolerance issues were more severe than their tinnitus, and more than 90% of veterans with PTSD specified unexpected impulse noises as those most likely to trigger challenging aversive responses.

Criterion F specifies the symptoms listed above must endure for more than one month.

The aforementioned phenomenon of auditory exclusion may result as an outcome related to the consolidation and recollection of traumatic memory [30,40]. In a variety of experiments, van Der Kolk and colleagues reported that study participants who were asked to recall and narrate events from the past did so in distinct ways, depending on whether events to be recalled were positive or traumatic. Narratives related to the traumatic events were “disorganized” as “subjects remembered some details all too clearly (…) but could not recall the sequence of events or other vital details…” (p.195 [40]). The lack of a coherent narrative displayed during the retelling of traumatic events was evident in both military [19] and civilian patients [20].

Diamond et al. [41] provided a model of emotional/traumatic memory consolidation that considered the long-known challenges related to accurate recall of traumatic events. The investigators detailed the time course through which limbic system structures, particularly the amygdala and hippocampus, were active during, and in response to, a traumatic exposure. Measures immediately following the exposure indicated both the amygdala and hippocampus were stimulated and active for up to several minutes; however, activity in the hippocampus ceased far sooner than activity in the amygdala. The investigators hypothesized that disorganization in trauma narratives could relate to the different elements of memory consolidated in the two limbic areas, with the hippocampus associated with narrative memory and the amygdala associated with emotional memory. The lack of coordination between the two areas during memory consolidation could interfere with a coherent and accurate recall of the event; this would contribute to the often disjointed and incomplete reports from patients. Summarizing this situation, van Der Kolk commented, “Traumatized people simultaneously remember too much and too little”. (p.181 [40]). Facilitating accurate recall of events surrounding a traumatic tinnitus onset might facilitate, in addition to coping with the event’s durable intrusiveness, management of symptoms reinforced by tinnitus as a reminder of the trauma.

## 3. Management

Moring et al. [32,42] assessed the post-traumatic effects as well as the possibility that intervention targeting PTSD symptoms might influence the severity of co-occurring conditions such as tinnitus and post-concussive headache. Symptoms of PTSD overlap considerably with a variety of other conditions whose presence and effects may substantially reduce a patient’s quality of life. As demonstrated in the Hinton studies [10,11], patients with traumatic exposures often experienced exacerbation of symptoms when confronted by trauma reminders. Such reminders also exacerbated co-occurring PTSD symptoms such as post-traumatic headache and hyperarousal. The finding that trauma and tinnitus could reinforce one another’s negative effects continued to challenge providers and patients; this relation was addressed by Moring et al., who conducted a pilot study [2] that revealed PTSD intervention had the potential to reduce tinnitus distress.

Their intervention centered on cognitive processing therapy, a front-line management strategy for patients with PTSD. Among the participants, the one-month intervention significantly recued the severity of PTSD symptoms. Tinnitus severity also declined, although not by a significant measure. However, the study demonstrated that management of one injury (PTSD) can support the coping for patients who also experience co-occurring tinnitus and depression. Their intervention supported the trauma-exposed individuals’ coping through counseling and cognitive processing therapy (CPT). The use of CPT for military personnel followed prior studies demonstrating durable positive effects for patients with PTSD. The technique supported a patient’s ability to process aversive symptom experiences in an accurate and reasonable manner; by doing so, patients may develop strategies and a workable lexicon that facilitates their navigating distressing memories and symptoms. As a consequence, the intervention may have improved patients’ abilities to distinguish effects of PTSD from those of tinnitus as it provided patients information and examples through which they could distinguish, and ultimately manage more effectively, the effects of both PTSD and tinnitus. The process, colloquially, could be considered one of “divide and conquer” through which a patient’s ability to manage myriad difficult symptoms improves as symptoms of one injury (i.e., PTSD) would be managed on their own, with decreasing contribution from symptoms associated with a different injury (i.e., tinnitus). Similar value related to facilitating management of the overlapping symptoms of hearing loss and tinnitus was described by Henry et al. [43].

If Janet’s suggestion that memories of traumatic events provide some of the fuel for their effects is correct, then management strategies targeting patients’ thoughts and beliefs regarding the event and its consequences might produce beneficial outcomes. One such strategy—the intervention demonstrating the strongest evidence for both—is cognitive behavioral therapy (CBT) [36,44]. Clinic Practice Guidelines from Europe [28] and the US [45] recommend CBT for subjective tinnitus ahead of sound therapy, hearing aid use, drugs, and other medical or alternative interventions. The American Psychiatric Association’s clinical guideline also provides CBT the strongest recommendation for management of PTSD [46]. 

The intervention is essentially noninvasive, collaborative, and patient-centered counseling, and results from large studies [45] show strong benefits for tinnitus management. McKenna et al.’s [9] cognitive model of tinnitus shares many attributes with PTSD models such as that provided by Schnurr & Janikowski [36]. The finding that CBT would reduce ratings of tinnitus severity to a greater extent than sound therapy suggests that the magnitude of non-auditory effects may be more strongly related to tinnitus severity than auditory effects. 

The notion of demystifying tinnitus [47] was long reported to have value to the patient, and investigators reported in the trauma literature many examples of educational counseling. A model describing the value of such counseling provided by Brewin [48] may be adapted for tinnitus. Brewin specified two forms of memory, verbally accessible memory (VAM) and situationally-accessible memory (SAM), corresponding to narrative/declarative memory and emotional/traumatic memory, respectively. Here, with consideration to Diamond et al. [41], the link between VAM and hippocampal-consolidated memory along with the link between SAM and amygdala-consolidated memory could provide one explanation for the disjointed memories associated with traumatic events. Brewin’s model went on to indicate that patients experience aversive responses when challenged by trauma reminders that activate emotional (SAM) memory. If patients can employ, or improve, narrative devices to recover and understand details, facts, and the features of the event (VAM), then they may more effectively manage the trauma reminder’s effects. When the patient’s narrative is thorough and accurate—in other words, when the VAM is enriched and available—the patient’s coping with trauma reminders should improve. Perhaps it is reasonable with regard to this model to substitute “tinnitus” for “trauma” and to recognize that counseling regarding tinnitus mechanisms and effects could improve a tinnitus-centered version of VAM, with effects for the patient similar to those described by Brewin.

CBT allows for counseling and dialogue with the patient to incorporate additional elements such as the therapeutic use of sound, relaxation exercises, mindfulness-based stress reduction, and sleep hygiene [44]. Despite the heterogeneity of both tinnitus and PTSD populations, there is potential for patients to find and hone management strategies that address diverse and often distressing long-term symptoms. Husain et al. [49] provided a glimpse of a mindfulness-based program on neuroanatomical structures through a small-sample pilot study. The investigators reported that the intervention was associated with changes in anterior and middle cingulate cortices, areas long-known as important areas for emotional processing [50]. Their small sample precluded firm conclusions; however, the possibility that a more thorough program of CBT, rigorously administered, could contribute to measurable changes in neuroanatomical structures would be a welcome addition to the practice of tinnitus management.

## 4. Conclusions

Although most people who report having tinnitus are not overly bothered by it, [1] those individuals who are bothered, often in combination with negative emotional states, or a diagnosed mental health condition, may evince catastrophic tinnitus effects. The consequences of trauma may produce a number of powerful symptoms that endure long after the trauma’s conclusion. When tinnitus and trauma are intertwined in the emotional state and life of the patient, any efforts at symptom management will be complicated and challenging due to the variety of powerful influences. In a way, the tinnitus clinician is fortunate; psychologists and trauma counselors pioneered many interventions and counseling strategies from the late 1800s to the present. In particular, CBT offers clinicians the opportunity to provide multi-disciplinary care focused on the specific needs and challenges expressed by patients. The co-occurrence of tinnitus and PTSD can exacerbate hyperarousal, impair sleep, and worsen the emotional state, yet many patients find some benefit from counseling that distinguishes the effects of tinnitus from those of PTSD. One such patient explained the relation between tinnitus and PTSD as that of an “…old married couple. They just get on each others’ nerves”. Future efforts intended to help such patients will need to account for the substantial mutually reinforcing effects that arise from the union of tinnitus and PTSD symptoms.

## Data Availability

Not applicable.

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
