# Peer review of "Tinnitus and Traumatic Memory"

_brainsci, 2022, doi:10.3390/brainsci12111585_

Round 1
Reviewer 1 Report
Congratulations on choosing this exciting but challenging topic. The manuscript reads very well, and the message conveyed by the Author is clear.
I have two suggestions:
The first suggestion is optional: please consider adding a concise paragraph dealing with auditory exclusion. This term is well recognized among police or military psychologists and relates to temporary changes in hearing caused by traumatic circumstances. For more details, please see https://scholar.google.com/scholar?hl=en&as_sdt=0%2C5&q=%22auditory+exclusion%22&btnG=
Auditory exclusion or auditory tunnel is often reported by persons (e.g., law enforcement officers or victims of traumatic experiences) under highly stressful or traumatic situations, often connected with the onset of PTSD.
The second suggestion is to discuss recent publications (see below), which would enrich and strengthen this paper.
Moring JC, Straud CL, Penzien DB, Resick PA, Peterson AL, Jaramillo CA, Eapen BC, McGeary CA, Mintz J, Litz BT, Young-McCaughan S, Keane TM, McGeary DD. PTSD symptoms and tinnitus severity: An analysis of veterans with posttraumatic headaches. Health Psychol. 2022 Mar;41(3):178-183. doi: 10.1037/hea0001113. PMID: 35298210.
Moring JC, Resick PA, Peterson AL, Husain FT, Esquivel C, Young-McCaughan S, Granato E, Fox PT; STRONG STAR Consortium. Treatment of Posttraumatic Stress Disorder Alleviates Tinnitus-Related Distress Among Veterans: A Pilot Study. Am J Audiol. 2022 Aug 24:1-6. doi: 10.1044/2022_AJA-21-00241. Epub ahead of print. PMID: 36001823.
Author Response
#1: Thank you for the careful reading and suggestions regarding additional items for manuscript inclusion. Regarding the specific suggestions:
Reviewer comments:
I have two suggestions:
The first suggestion is optional: please consider adding a concise paragraph dealing with auditory exclusion. This term is well recognized among police or military psychologists and relates to temporary changes in hearing caused by traumatic circumstances. For more details, please see https://scholar.google.com/scholar?hl=en&as_sdt=0%2C5&q=%22auditory+exclusion%22&btnG=
Auditory exclusion or auditory tunnel is often reported by persons (e.g., law enforcement officers or victims of traumatic experiences) under highly stressful or traumatic situations, often connected with the onset of PTSD.
Response 1. Auditory exclusion and related references (including one mentioning visual exclusion) now appear in the “Shared Mechanisms: Hearing” section. The topic area is related to the information provided in this MS and its inclusion provides another example of the consequences of traumatic exposures. Thank you for the suggestion.
The second suggestion is to discuss recent publications (see below), which would enrich and strengthen this paper.
Moring JC, Straud CL, Penzien DB, Resick PA, Peterson AL, Jaramillo CA, Eapen BC, McGeary CA, Mintz J, Litz BT, Young-McCaughan S, Keane TM, McGeary DD. PTSD symptoms and tinnitus severity: An analysis of veterans with posttraumatic headaches. Health Psychol. 2022 Mar;41(3):178-183. doi: 10.1037/hea0001113. PMID: 35298210.
Moring JC, Resick PA, Peterson AL, Husain FT, Esquivel C, Young-McCaughan S, Granato E, Fox PT; STRONG STAR Consortium. Treatment of Posttraumatic Stress Disorder Alleviates Tinnitus-Related Distress Among Veterans: A Pilot Study. Am J Audiol. 2022 Aug 24:1-6. doi: 10.1044/2022_AJA-21-00241. Epub ahead of print. PMID: 36001823.
Response 2. I have included both Moring et al references. The first (although it’s the second listed by the reviewer) appears in the abstract as an element to consider for future investigation of this topic. The possibility that intervention for PTSD could ameliorate tinnitus symptoms is referred to in the management section of the MS as well. The second Moring et al reference is alluded to in the PTSD section as an indicator of symptoms other than those specified in the DSM. The interactions between symptoms of tinnitus and PTSD likely cannot be overstated, and the inclusion of the Moring studies (esp. as they are new), improves the MS considerably. Thank you for these suggestions, and again for the review.
Reviewer 2 Report
Dear Ladies and Gentlemen, Dear Journal-Team,
the manuscript 'Tinnitus and traumatic memory' describes the relations and interactions of tinnitus and posttraumatic stress disorder and underlines the importance that emotions and brainprocessing have for sensory perception. It is well written. A table where all associations and relations between tinnitus and posttraumatic stress disorder are summarized would be helpful for understanding. Check the references for accuracy according to the Journal Style Guidelines, and check references 5, 9 and 36 for accuracy.
Sincerely,
Author Response
Reviewer 2 comments:
the manuscript 'Tinnitus and traumatic memory' describes the relations and interactions of tinnitus and posttraumatic stress disorder and underlines the importance that emotions and brain processing have for sensory perception. It is well written. A table where all associations and relations between tinnitus and posttraumatic stress disorder are summarized would be helpful for understanding. Check the references for accuracy according to the Journal Style Guidelines, and check references 5, 9 and 36 for accuracy.
Response: Thank you for the careful reading and for the suggestion regarding a table and the references. I am hopeful that the reference section is now accurate (several references have been added as well). Regarding a table to serve as a summary, I admit to a substantial struggle with this addition. I agree that a table to summarize is a good idea, however my attempts, thus far, produced a large cumbersome table that likely takes up more space than it is worth. If this is an item held as a priority by the reviewer, then I will continue the effort, however I thought it prudent to submit now. Again, while I agree a table could be helpful, in practice, I struggled to generate one. I will try again if that is the reviewer’s preference. Thank you for understanding.
Reviewer 3 Report
This is a very interesting review linking the phenomena of tinitus with traumatic memory, particularly with PTSD. The paper is of interest for the journal and the readers; however I recommend several changes to be made .
Abstract
1- Before describing, what the review is aiming to contribute, I recommend to present a brief introduction of the topic: tinnitus and traumatic memory.
2- The aims and methods of the review should be described in the abstract section. If it is a narrative review, the authors should also describe how they conducted it.
3- Some recommendations for future studies would be recommended (just in one sentence) in the abstract section.
Introduction:
1- The introduction section could be divided into several subsections. Tinnitus can be the first. I recommend to describe some other causes for tinnitus rather than mental health status. AT the beginning of the introduction, for instance...
2- Tinnitus and trauma can be a second subsection (Line 54).
3- PTSD, concept and prevalence, can be the third subsection of the introduction. What does PTSD mean irrespective of the definition of the DSM?
Methods
1- A methods section is necessary to understand how the authors selected the references to be included in their review.
PLease, describe how the papers were selected (inclusio/exclusion criteria). There were any temporal criteria for including them? Search terms?
Management: CBT
1- A shared therapy for two disorders is not enough to link these two disorders. PLease, rephrase the first sentence of this section (Management: CBT). It would be better to say that CBT is effective for both. Please, add some more references.
Funding/Acknowledgments and conflicts of interest: this information is lacking.
Author Response
Reviewer 3 comments:
Abstract
1- Before describing, what the review is aiming to contribute, I recommend to present a brief introduction of the topic: tinnitus and traumatic memory.
Response 1: The introduction and abstract were re-written, the introduction considerably. I added reference to Diamond et al’s model of traumatic memory in the Shared Mechanisms: Psychological section with the intent to elucidate more clearly the relation between tinnitus and traumatic memory. Thank you for this suggestion.
2- The aims and methods of the review should be described in the abstract section. If it is a narrative review, the authors should also describe how they conducted it.
Response 2: The aim of the review is now presented more clearly in the abstract and introduction. I added a brief methods section; as this is offering is not a systematic review, I fear that I may not be complying with the requests of reviewer #3, and I regret any shortcoming on my part. I endeavored to make the aims clear, and to admit in a forthright manner the justification for including the literature considered herein.
3- Some recommendations for future studies would be recommended (just in one sentence) in the abstract section.
Response 3: Thank you for this suggestion, the addition of the Moring et al reference and its reiteration later in the MS were intended to address this item.
Introduction:
1- The introduction section could be divided into several subsections. Tinnitus can be the first. I recommend to describe some other causes for tinnitus rather than mental health status. AT the beginning of the introduction, for instance...
Response 1: I have divided the introduction accordingly, and moved several passages to points later in the text in order to keep the introduction to a reasonable length. This suggestion substantially improved the organization of the MS; thank you.
2- Tinnitus and trauma can be a second subsection (Line 54).
Response 2: done
3- PTSD, concept and prevalence, can be the third subsection of the introduction. What does PTSD mean irrespective of the definition of the DSM?
Response 3: done
Methods
1- A methods section is necessary to understand how the authors selected the references to be included in their review. PLease, describe how the papers were selected (inclusio/exclusion criteria). There were any temporal criteria for including them? Search terms?
Response 1: as indicated above, a brief section was added, although the source material reviewed herein was not accessed using search items, etc.
Management: CBT
1- A shared therapy for two disorders is not enough to link these two disorders. PLease, rephrase the first sentence of this section (Management: CBT). It would be better to say that CBT is effective for both. Please, add some more references.
Response 1: Thank you for this comment. I agree that this section required more references and clarifying. The management section was overhauled considerably with the reviewer comments in mind. The Moring et al work is now referenced in this section, as is reference to a model provided by Brewin (2002) that addresses specifically trauma’s effects on memory, and the value of counseling to improve a patient’s ability to manage responses in challenging environments.
Thank you again for the careful reading and suggestions.
Funding/Acknowledgments and conflicts of interest: this information is lacking.
Response: added at the end of the MS